# Explorative Latent Self-Supervised Active Search Algorithm (ELSA)

## Abstract

In computer vision, attaining exceptional performance often necessitates access to large labeled datasets. The creation of extensive datasets through manual annotation is not only cost-prohibitive but also practically infeasible due to the scarcity of positive samples in imbalanced datasets where negative samples dominate. To tackle this intricate problem, we introduce Efficient Latent Space-based Self-Supervised Active Learning Search (ELSA), an active learning-based labeling assistant. ELSA distinguishes itself from existing interactive annotation methods by focusing exclusively on positive class labeling in massively imbalanced datasets replete with a substantial number of negative samples. Through the automatic exclusion of the majority of negative samples, ELSA achieves a remarkable level of precision and accuracy in its search. This novel framework comprises three fundamental components: a)an iterative Nearest Neighbor Search, b)a Sophisticated Random Sampler, c)a Linear Head powered by Active Learning. Our comprehensive study provides insights into the interplay of these components and their collective impact on search efficiency. Notably, we demonstrate that ELSA achieves orders of magnitude superior performance, in average starting with as little as 5 or less positive samples in ImageNet 1k we managed to detect as much as 80% of all the examples belonging to that class by only labeling as little as 0.67% of the entire dataset manually.

## 1 Introduction

Supervised Learning algorithms(He et al. (2015), Liu et al. (2021),Dosovitskiy et al. (2021), Liu et al. (2022)) in computer vision have shown impressive results in tasks of classification, detection, segmentation, pose-estimation etc. One drawback of self-supervised learning is the requirement of large number of labelled samples in data. This problem is partially solved by creating huge datasets such as CIFAR[1], ImageNet[2], COCO[3] etc., to deal with specific tasks such as classification, detection, segmentation etc. The creation of such datasets is often time-consuming and labour-intensive. Although well-labelled dataset exists for most day-to-day vision-based tasks, but when it comes to new novel applications(Medical science, Industry etc.) the dataset needs to be created and labelled by a human annotator. There have been works to automate the tasks such as segmentation(Sourati et al. (2019),Top et al. (2011)), pose estimation(Feng et al. (2023)), video classification(Yan et al. (2003)), speech(Desmond et al. (2021),Zhang et al. (2015)) etc.

In this work, we deal with the task of classification or labelling the images of a specific class in a large unlabelled dataset.There are works that try(Desmond et al. (2021),Zhang et al. (2015),Benato et al. (2021),noa,Sourati et al. (2019),Feng et al. (2023),Yan et al. (2003),Top et al. (2011)) to automate the task of annotation by an AI assistant for classification but most of these works either focus on labelling the entire dataset or try to label those samples that make the classification head achieve the best performance in a binary classification setting. This work is fundamentally different since we are considering real-life datasets where we have huge number of negative samples compared to positive one and we are interested in finding only the positive ones with minimal human effort.

---

[1] https://www.cs.toronto.edu/~kriz/cifar.html
[2] https://www.image-net.org/
[3] https://cocodataset.org/

In our work, we engage with two fundamental challenges. Firstly, we aspire to develop a system capable of arranging objects with similar characteristics closely together within a specific space. Secondly, once we've sorted these objects into groups, we need to locate and identify multiple instances of these groupings within the larger space.

To tackle the first challenge, we enlist the assistance of intelligent computer programs such as VI-CReg(Bardes et al. (2022)) ,BYOL(Grill et al. (2020)). These programs function like skilled detectives, examining the characteristics and features of the objects we're interested in and then placing them into categories based on their shared traits. Essentially, they are helping us create order and structure within our data.

For the second challenge, we employ two distinct methods:

- Nearest Neighbor Search (NN): Imagine that objects with similar characteristics are like neighbors in a big neighborhood. The Nearest Neighbor Search method behaves like a diligent explorer who repeatedly checks the houses in the neighborhood to find all the ones that match our description. It keeps searching until it is confident that it has found them all.

- Random Search (RandS): The positive samples might be distributed in the form of multiple separate clusters in the feature space separated by a large number of negative samples, in which case NN can never find those samples(located at a different cluster). To solve this problem, we use Random Search(RandS) to find random clusters from the latent space most likely made of positive samples based on the region of disagreement calculated by our active learning algorithm.

To solve the problem we are proposing ELSA: Efficient Latent Space-based Self-Supervised Active Learning Search. We have used ImageNet 1k to demonstrate the working of our algorithm.[4].ELSA, is composed of three main components, A Nearest Neighbour search module, A Random Sampler and a classification head. The three components work together to make the process of annotation orders of magnitude more efficient.

## 2 RELATED WORKS:

**Self Supervised Learning:** Self-Supervised learning focuses on the production of information-rich feature vector using no labels. This can be achieved in a number of ways. In contrastive method similar types of examples are brought closer to each other while dissimilar types are as far as possible from each other. These methods include SwAv(Caron et al. (2019)), MoCo(Chen et al. (2020b)), SimCLR(Chen et al. (2020a)), VICReg(Bardes et al. (2022)),BarlowTwins(Zbontar et al. (2021)),(Bromley et al.). Self Supervised Learning also includes other methods like clustering(Xie et al. (2017),Bautista et al. (2016), Huang et al. (2019), Yang et al. (2016)), Distillation (BYOLGrill et al. (2020)),ObOw(Gidaris et al. (2021)),SimSiam(Chen & He (2020)) etc.

**Interactive Labeling:** There have been attempts in solving the labeling problem using an AI assistant (Desmond et al. (2021)).It tries to label data along with the user and the performances are tracked for the agent for every iteration. As new labels are learned, the agent improves. When the performance of the agent is on par with expectations the remaining dataset is entirely labeled by the agent (Zhang et al. (2015)).It tries to solve the problem of labeling emotions in speed data using active and semi-supervised learning (Benato et al. (2021)), to use feature space projections to annotate the labels in an active learning setting (noa), to find important information in the documents to make the classification easier for the user(Yan et al. (2003), Feng et al. (2023)), auto annotate video and 3D pose estimation data using active learning. While Sourati et al. (2019), Top et al. (2011) tries to automate the task of segmentation.

**Active Learning**: Active Learning(AL) is a paradigm where a computer and a human/oracle work in unison to achieve large performance gain in tasks by using as little labeled data as possible. The way each unlabeled sample is queried broadly divides AL framework into three main divisions, Membership Query Synthesis (Angluin (1988),King et al. (2004)) where the learner can ask the oracle to label any sample from the input space even the ones generated by it, stream-based selective sampling (Dagan & Engelson (1995),Krishnamurthy (2002)) and pool-based (Lewis & Gale (1994))

---

[4]https://www.image-net.org/

where the learner asks the oracle to label data from a predefined set of unlabeled data differing only in the fact that the former makes the querying sequentially while the later does it in a batch. Our work is an example of pool-based AL. The combination of Deep Learning(DL) and Active Learning(AL), DeepAL has been used extensively in computer vision tasks such as Image Classification and Recognistion (Folmsbee et al. (2018),Du et al. (2018),Bukala et al. (2020),Alahmari et al. (2019),Stark et al. (2015)), Object Detection and Segmentation (Roy et al. (2019),Norouzzadeh et al. (2019),Kellenberger et al. (2019),Feng et al. (2019)), Video(Hussein et al. (2016),Hossain et al. (2018),Wang et al. (2018),Aghdam et al. (2019)).

## 3 DEFINITIONS AND PRELIMINARIES

Before moving on to the algorithms we formalise the problem statement a bit more. The following sections give a brief understanding of the problem, the metrics involved and some functions that are used.

### 3.1 DATASET, LABELLING EFFICIENCY & DISCOVERY RATE

Consider a dataset $\mathcal{D}$, which has a large number of images $I_i \in \mathcal{D}$. The self-supervised model $g$, maps each of these datapoints to a feature vector $z_i \in \mathcal{R}^n$.

$$g : \mathcal{D} \mapsto \mathcal{R}^n \tag{1}$$

We define:

$$\mathcal{D}^+ : \text{set of all positive samples} \tag{2}$$

$$\mathcal{D}^- : \text{set of all negative samples} \tag{3}$$

Let $\mathcal{L}$ be a set of images such that $\mathcal{L} \subseteq \mathcal{D}$. We label the set $\mathcal{L}$, and find that $L = L^+ \cup L^-$, where $L^+$ corresponds to the set of positive samples and $L^-$ corresponds to the set of negative samples. We define seeds as the few initial positive samples/query that we begin our search with.

$$Discovery\ Rate(d_r) = \frac{||L^+||}{||D^+||} \tag{4}$$

$$Labelling\ Efficiency(L_e) = \frac{||L^+||}{||L||} \tag{5}$$

- The Discovery Rate, denoted as $d_r$ is characterized as the quotient of positive samples identified by the algorithm compared to the total positive samples encompassed within the dataset.
- Labeling Efficiency, denoted as $L_e$ is characterised as the quotient of positive samples identified by the algorithm compared to the total number of samples labelled during the search process.
- In ideal cases, i.e., an ideal algorithm $d_r = 1$, and $L_e = 1$.
- In the case of a untrained guessor (i.e. an untrained model), the discovery rate is unity, then labelling efficiency $(L_e) = \frac{||D^+||}{||D||}$, which for ImageNet dataset is nearly $0.1\%$.
- Labelling efficiency and discovery rate has a negative correlation. Increasing the reach radius increases the discovery but reduces labelling efficiency.

### 3.2 SELF SUPERVISED MODELS, EMBEDDING SPACE & PROJECTION SPACE

Most of the state of the art, self-supervised architectures have joint embedding architecture where, a feature extractor backbone, converts the image into a feature vector based mebedding space, which is later fed into a projector network, which generally projects the produced feature vector into a higher dimensional feature vector. The self-supervised architecture is made up of the following components.

Given a batch of Images $I$, we define the encoder $f_{enc}$ and $f_{proj}$ to be:

$$f_{enc} : I \mapsto Y; Y \subseteq \mathbb{R}^n \tag{6}$$

$$f_{proj} : Y \mapsto Z; Z \subseteq \mathbb{R}^n \tag{7}$$

### 3.3 NEAREST NEIGHBOUR SEARCH

Most self-supervised algorithms work on the principle of contrastive learning, making similar samples to lie in clusters. Intuitively, one might believe that the Nearest Neighbour search should work well given that the positive samples form a cluster in the latent space. However, this is not true due to the following reasons:

1. **Sparse Clusters**: In scenarios where clusters are loosely packed and contain various negative samples, labeling efficiency significantly deteriorates. This issue primarily stems from limitations in the self-supervised architecture, which fails to segregate samples into cohesive clusters. Consequently, the nearest neighbor search algorithm struggles to efficiently discover positive samples.

2. **Multiple Disconnected Clusters**: Latent spaces may contain positive samples divided into several separate clusters, which may or may not be interconnected. In such cases, the nearest neighbor search is unable to identify samples from different clusters, hampering its utility.

3. **Boundary Queries**: Queries placed on the boundary of a cluster, in a latent space of dimension N, necessitate a doubled search radius. This arises from the scaling behavior of the volume in high-dimensional spaces ($r^N$). Consequently, as we move away from the cluster center, the search efficiency experiences an exponential decline. The sample density (number of samples within a unit latent space volume), approximated as a constant, exacerbates this problem.

4. **Out-of-Cluster Queries**: If a query sample lies outside any cluster, nearest neighbor search utterly fails. Moreover, depending on the query's distance from the cluster center, the computational burden of finding similar samples grows exponentially. This situation represents an extreme case and is typically attributed to the query's location, making it exceptionally challenging to handle.

Addressing the first three of the aforementioned challenges, our proposed solution, Efficient Latent Space-based Self-Supervised Active Learning Search (ELSA), offers a remedy. The RandS component of ELSA manages the first two problems, while the NN component mitigates the third one since ELSA is an iterative algorithm. The fourth problem can be solved by taking multiple seeds/queries in the beginning.

## 4 ELSA: ARCHITECTURE AND DESCRIPTION

The algorithm is composed of multiple elements that all work together. Each of which is described below

- Nearest Neighbour Search(NN): A nearest neighbour search module that works with multiple queries details of which is described in section 4.1
- Sampler: This is an active learning-based sampler that assists the RandS component. It tries to find the most likely positive samples that are not yet discovered.
- Random Search (RandS): The algorithm for this component is described in section 4.2. The RandS component receives a list of the most likely positive samples from the sampler. For every sample received, a NN search is done on the dataset to find the probable cluster centre to which these samples might belong. RandS returns a random subset from this list of cluster centres.
- Oracle: Since our algorithm is motivated from Active Learning, we have to define the Oracle, a human labeller that labels the sample after every step. For our experiments, we have used the pre-assigned labels from the ImageNet dataset.
- Encoder(Feature Extractor): A CNN-based (ResNet-50) encoder trained by self-supervised learning encodes the image into low-dimensional encoding vectors.
- Projector: A MLP block that projects the encoding vectors to latent space.
- Final Search: ELSA being an iterative algorithm trains the classification head after every epoch. Once all the iterations are over the sampler is used one last time to predict the most likely positive samples that are not yet discovered,

## 4.1 NEAREST NEIGHBOUR SEARCH(NN)

In the embedding space we now define:

$\Lambda$ : Set of R samples/queries.
The set $\Lambda$, is essentially the set of samples from the positive class, whose members we want to discover.

---

**Algorithm 1** Nearest Neighbour

    **Input**: $\Lambda$, a                                       $\triangleright$ a is a hyperparameter
,     **Output**: $\delta$                      $\triangleright$ Nearest Neighbours corresponding to the given queries
  1: $g(x) = f_{proj}(f_{enc}(x))$                            $\triangleright$ Defining the function g(x)
  2: $\Lambda \subseteq \mathcal{D}$, $a$ is a hyperparameter such that $a \in \mathbb{Z}^+$. $\mathcal{M}_{||\mathcal{D}||\times||\Lambda||}$, such that $\mathcal{M}_{ij} = MSE(g(d_i)), g(\Lambda_i)); d_i \in D, \Lambda_i \in \Lambda$.
  3: $\mathcal{R}_i = \arg\min_j (\bigcup M_{ij})$ ; $\min_j(\bigcup M_{ij}) \geq 0$. $R = \bigcup_i R_i$, and we have then $||R|| \leq ||D||$.
  4: $\mathcal{R}^* = Sort_{ascending}(\mathcal{R})$; sorting in accordance with $Min(M_{ij})|j \in ||\Lambda||$
  5: $\delta$ is then the first $a$ samples of $R^*$.
  6: Return $\delta$

---

## 4.2 RANDOM SEARCH (RANDS)

We define $L$ as a subset of $\mathcal{D}$ such that $L \subseteq \mathcal{D}$; $b \in \mathbb{Z}^+$ and $l \in \mathbb{Z}^+$. RandS is different from the usual textbook random search algorithm mainly because it finds probable positive clusters rather than positive data points. To achieve this, a nearest neighbour search component is used inside the RanS sampler. The main motivation behind doing this is to keep the two components of the search algorithm perpendicular to one another. The cluster centres found by the RandS component is used by the NN component to find all the samples from that specific cluster. The random search component of the algorithm can be described as follows:

---

**Algorithm 2** Random Search Algorithm

    **Input**: $L*$, $b$
    **Output**: $\delta$
  1: Take $L^*$ to be a set of randomly sampled points of cardinality $l$ from $L$
  2: $b \leftarrow$ Number of random samples
  3: $\delta = NN(L^*, b)$                     $\triangleright$ NN is the Nearest Neighbour Algorithm
  4: Return $\delta$

---

## 4.3 SAMPLER

We use a simple estimator, which can be replaced by more sophisticated estimators later in the future. The sampler algorithm is described in algorithm 3.

---

**Algorithm 3** Sampler Algorithm

    **Input**: $D$
    **Output**: $S_i$
  1: $Y_i \leftarrow f_{enc}(I_i)$
  2: $Z_i \leftarrow f_{proj}(Y_i)$
  3: $S_i \leftarrow \text{Head}(Z_i)$
  4: $H = S_i, I_i \in D$
  5: $T = \text{Mean}(H) + \alpha\text{Std}(H); \alpha \in \mathcal{R}$
  6: Samples $= \{S_i|S_i > T; I_i \in D\}$

---

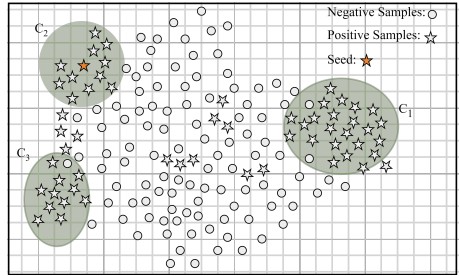

Figure 1: Working of ELSA

## 4.4 WHY ELSA WORKS

Considering a simple toy problem with latent dimension two and a single a single-seed(positive example) and a set of unlabelled data points as denoted in figure 1. Now we the task is to find all the positive samples from that single given seed. Most of the positve samples lie in either cluster $C_1$, $C_2$ or $C_3$. If we do a simple nearest neighbour search, we will successfully discover all the samples in $C_2$ but fail to reach $C_1$ and $C_3$ unless the search radius is large. This is the boundary query problem described in section 3.3. Using an iterative nearest neighbour search solves this problem. Still, the cluster $C_1$ remains undiscovered. To solve this problem, we use RandS. Using the knowledge gained from training the head on elements from cluster $C_3$ and $C_2$, the model can discover the cluster $C_3$. Still, certain points remain undiscovered that don't form clusters or are in sparse clusters. To solve this problem, we do the final search as described in section 4.

## 4.5 ELSA: ARCHITECTURE

The ELSA combines the Nearest Neighbour search algorithm with RandS guided by the active learning-based sampler to discover as many positive samples as possible using the least amount of human annotation effort. The algorithm is described in algorithm 4.

---

**Algorithm 4** ELSA Algorithm

---

    **Input**: $\Lambda$         ▷ Initial seeds/queries
    **Output**: $\Lambda^*$         ▷ Updated value
1:  $\delta^0 \leftarrow \Lambda$         ▷ Intializing $\delta^0$ with the value of $\Lambda$
2: **for** i $\leq$ number of epochs **do**
3:     $K \leftarrow NN(\delta^0, a)$         ▷ a is a hyperparameter
4:     $l \leftarrow \text{Sampler}(\mathcal{D})$
5:     $\delta \leftarrow \text{RandS}(l, b)$         ▷ b is a hyperparameter
6:     $\gamma \leftarrow K \bigcup \delta$
7:     Oracle$(\gamma) \rightarrow (\gamma^+, \gamma^-)$         ▷ Partioning into positive and negative samples
8:     $\Lambda \leftarrow \Lambda \bigcup \gamma^+$
9:     Header trained on $(\Lambda \bigcup \gamma^-)$
10:    $\delta^0 = \gamma^+$
11:    $i + +$
12: **end for**
13: $z \leftarrow Sampler(D)$
14: $Oracle(z) \rightarrow (z^+, z^-)$         ▷ Oracle labels as positive or negative samples
15: $\Lambda^* \leftarrow \Lambda \bigcup Z^+$; Return $\Lambda^*$

---

## 5 SPACE AND TIME COMPLEXITY ANALYSIS

**Time Complexity**
We shall do a step by step move to calculate the time complexity. We take $d$ to be the time taken for calculating the MSE distance in the latent space, and $H = O(MLP(z_i))$, where $z_i$ is the $i^{th}$ datapoint.

- **Nearest Neighbour**: Considering there are $k$ queries and $n$ points in the dataset. To calculate the nearest neighbours, it will take $O(ndk)$ time. To sort the list and pick the top $a$ samples it takes $O(n \log n)$. Thus we have then $O(ndk + n \log n) \approx O(ndk)$

- **Sampler**: The sampler scans through the entire dataset to calculate the region of disagreement, and takes $O(nH)$ time.

- **Random Search**: Considering $l$ random seeds from the sampler. To calculate the nearest neighbour $O(ndl)$. To sort the lines and take the $b$ samples it takes roughly $O(n \log n + ndl) \approx O(ndl)$

Thus the total time complexity $O(ndk + ndl + nH) \approx O(nC)$.

**Space Complexity**

The evaluation of space complexity is rather simple in this case. We need to store matrices of dimensions $n \times l$ when performing the searches, and it is the heaviest part of the computation. Regarding space, the complexity comes out to be of $O(nl)$, where $n$ is the dataset size and $l$ is the number of queries.

## 6 RESULTS

We have taken 50 randomly sampled classes from the ImageNet 1K dataset to test our algorithm. For all the experiments we fix the number of epochs to 10, we set a and l to 50, and b is initially 50 but increases linearly as more positive samples are discovered. We stip away various components from ELSA to test the effectiveness of each component which leads us to the following cases (refer to table 1):

- ELSA (NN + RandS + Active Learning + Final Search)
- No-Active (NN+ RandS + Final Search)
- Nearest Neighbour (NN)

| Algorithm | INN | Random Sampler | Active Learning | Final Search |
|:---:|:---:|:---:|:---:|:---:|
| ELSA | ✓ | ✓ | ✓ | ✓ |
| No Active | ✓ | ✓ | ✗ | ✓ |
| Nearest Neighbour | ✓ | ✗ | ✗ | ✗ |

Table 1: The different algorithms used for comparison

As discussed earlier, our algorithm can be evaluated on the basis of two interlinked parameters. The first one being the Discovery rate ($d_r$), and the second parameter is the labelling efficiency ($L_e$) (refer to section 3.1). These parameters have strong negative correlation among each other. We are interested in maximizing both the parameters simultaneously and optimize. To make a just comparison we will state the performance of the algorithm in terms of both parameters.

### 6.1 ELSA: ANALYSIS AND RESULTS

To quantify the performance of our algorithm, we have used to separate self-supervised algorithms to build our projector and feature space. We are using Barlow Twins and VICReg. The choice of these two specific algorithms can be justified on the fact that both of them have the same architecture for the embeddign and latent spaces and are more or less the SOTA in self-supervised learning.

We define the number of seeds as the number of queries with which we begin our search with. All values are median values, and are reported in table 2. We observe that the model's performance increases as the number of seeds increase. This is because multiple seeds give multiple views of the query, and hence, the algorithm can find the desired samples more effectively.

### 6.2 COMPONENT WISE ANALYSIS OF ELSA

The results of table 2, clearly depicts the importance of various components of ELSA. Using only the NN component gives us a good labeling efficiency but when it comes to discovery, it performs

| No of. Seeds | VICReg | | Barlow and Twin | |
|---|---|---|---|---|
| | Labelling Efficiency | Discovery Rate | Labelling Efficiency | Discovery Rate |
| 1 | 0.093 | 0.6192 | 0.046 | 0.706 |
| 2 | 0.107 | 0.7046 | 0.055 | 0.786 |
| 5 | 0.115 | 0.7823 | 0.054 | 0.832 |
| 10 | 0.117 | 0.7807 | 0.057 | 0.855 |
| 50 | 0.086 | 0.8768 | 0.041 | 0.932 |
| Median | 0.1061 | 0.798 | 0.050 | 0.860 |

Table 2: The discovery and labelling percentages of ELSA on VICReg and Barlow Twins latent space

| | VICReg | | | | | | | |
|---|---|---|---|---|---|---|---|---|
| No of seeds | ELSA | | No Active | | Nearest Neighbour | | NN_Large | |
| | Labelling Efficiency | Discovery | Labeling Efficiency | Discovery | Labeling Efficiency | Discovery | Labeling Efficiency | Discovery |
| 1 | 0.093 | 0.6192 | 0.0163 | 0.51 | 0.15 | 0.0323 | 0.1233 | 0.0369 |
| 2 | 0.107 | 0.7046 | 0.0186 | 0.66 | 0.15 | 0.0331 | 0.1461 | 0.0573 |
| 5 | 0.116 | 0.7823 | 0.0218 | 0.81 | 0.24 | 0.0761 | 0.1707 | 0.0723 |
| 10 | 0.117 | 0.7807 | 0.0211 | 0.84 | 0.26 | 0.0661 | 0.1786 | 0.1046 |
| 50 | 0.086 | 0.8768 | 0.0177 | 0.92 | 0.41 | 0.1173 | 0.2866 | 0.1176 |

Table 3: This table encapsulates the component wise analysis of ELSA with VICReg.

| | Barlow Twins | | | | | | | |
|---|---|---|---|---|---|---|---|---|
| No of seeds | ELSA | | No Active | | NN | | NN Large | |
| | Labeling Efficiency | Discovery | Labeling Efficiency | Discovery | Labeling Efficiency | Discovery | Labeling Effciency | Discovery |
| 1 | 0.046 | 0.7061 | 0.0167 | 0.706 | 0.148 | 0.0346 | 0.1147 | 0.0396 |
| 2 | 0.055 | 0.7861 | 0.0185 | 0.767 | 0.194 | 0.0471 | 0.1149 | 0.0531 |
| 5 | 0.054 | 0.8323 | 0.0199 | 0.846 | 0.248 | 0.0846 | 0.1900 | 0.0923 |
| 10 | 0.057 | 0.8554 | 0.0187 | 0.867 | 0.345 | 0.0907 | 0.1955 | 0.0838 |
| 50 | 0.042 | 0.9323 | 0.0181 | 0.935 | 0.412 | 0.0984 | 0.3266 | 0.1488 |

Table 4: This table encapsulates the component wise performance of ELSA with Barlow Twins

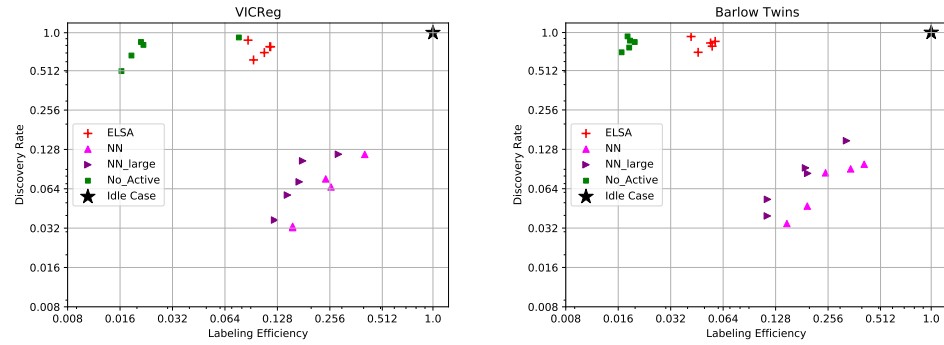

Figure 2: Component-wise analysis of ELSA with VICReg and Barlow Twins

poorly. Not using the active learning sampler makes the very inefficient in terms of labelling efficiency. This phenomenon can be visualised by figure 2

## 6.3 ELSA ON EMBEDDING SPACE

Till now we were testing our algorithm on the latent space of VICReg and Barlow Twins. Now we see if using the embedding space in place to of the latent space affects our algorithm's performance. We train for the same 50 classes and similar experiment structure that we used in (tables 2, 3, 4). We take median statistics and report our finding in 5

| Algorithm | Discovery | Labelling Efficiency | Linear Evaluation Top 1 |
|---|---|---|---|
| **VICReg** (Bardes et al. (2022)) | 0.78 | 0.134 | 73.2 |
| **Barlow Twins** (Zbontar et al. (2021)) | 0.83 | 0.061 | 73.2 |
| **MoCo** (He et al. (2020)) | 0.69 | 0.017 | 60.6 |
| **MoCo_V2** (Chen et al. (2020b)) | 0.88 | 0.026 | 71.1 |
| **ObOW** (Gidaris et al. (2021)) | 0.89 | 0.051 | 73.8 |
| **SimSiam** (Chen & He (2020)) | 0.67 | 0.035 | 71.3 |
| **SwAV** (Caron et al. (2019)) | 0.83 | 0.032 | 71.8 |
| **SwAV-multi-crop** (Caron et al. (2019)) | 0.83 | 0.071 | 75.3 |

Table 5: Performance of ELSA on different Embedding spaces

### 6.4 DISCOVERY VS. EFFICIENCY GRAPH FOR THE THREE ALGORITHMS ON VICREG

In an ideal scenario, a perfect algorithm would attain both a discovery rate and efficiency equal to unity. To assess our algorithm's performance relative to others, we construct a graph that plots discovery rate against labelling efficiency across various experimental runs. We determine the corresponding labelling efficiency and discovery rate by varying hyperparameter values. Figure 3 presents three specific instances. In all cases, ELSA consistently outperforms a basic nearest neighbor search.

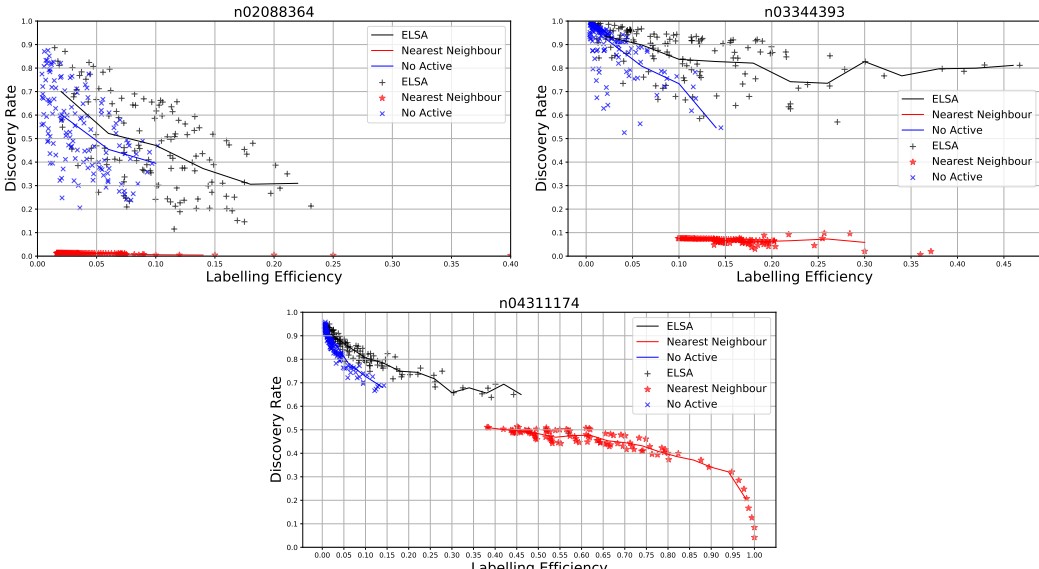

Figure 3: Cherry picked classes with n04311174; tight cluster where all three algorithms perform well, n02088364; sparse cluster where all the three algorithms perform poorly, n03344393 typical example where ELSA overpowers NN

## 7 CONCLUSION

In this work, we introduce ELSA to alleviate the task of labelling large datasets. From our analysis, we have shown how ELSA can help reduce the time required to label samples by orders of magnitude compared to manual labelling in large datasets dominated by many negative samples. With ELSA, we can label as much as 80 % of all positive classes by only labelling as little as 0.67 % of the dataset using five or less initial queries. In our future work, we aim to enhance the performance of the active learning sampler by focusing on identifying shared traits for querying. We would also like to extend the scope of the labeling across other categories of datasets such as text, voice and other multimodal datasets as well.

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

## A  APPENDIX: WORKING OF ELSA

In section 4.4 we briefly described the working of ELSA and the intuition behind using the three components as described in section 4. We now present a detailed discussion on the working of ELSA. The components of ELSA work symbiotically. For better understanding we provide a flow chart in figure 4 depicting the working of ELSA. Before moving forward, we reiterate that ELSA

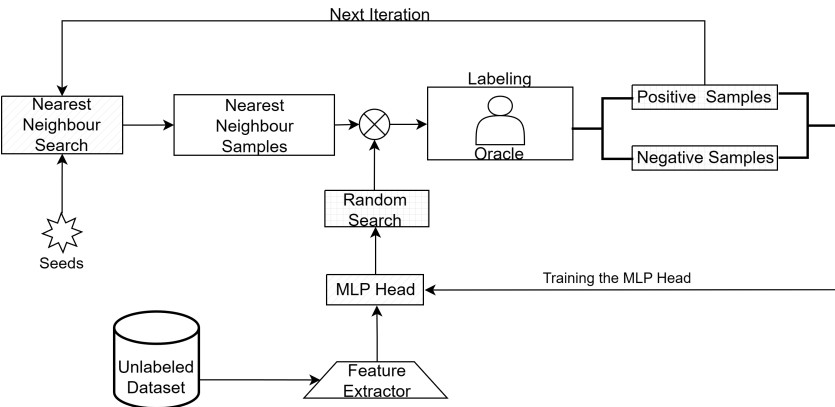

Figure 4: Architecture of ELSA

works on the projection vectors produced by self-supervised algorithms. The distance between any two points or samples is simply the MSE distance between their projection vectors. Now, following settings similar to the ones introduced in section 4.4 we try to understand how and why ELSA works. In this example, we have considered four clusters $C_1$, $C_2$, $C_3$ and $C_4$ where $C_2$ and $C_3$ are connected to each other by positive samples. The working of the algorithm is visualised in figure 5. The positive samples discovered after every step is used to train the active learning head after every step.

**Iteration 1** On the very first iteration, only a single point is discovered, which is also our initial seed. We aim to discover the remaining positive samples only using this single seed.

**Iteration 2** After the end of the first iteration, we see that the algorithm has managed to discover the majority of points in $C_2$. All these points lie in the neighbourhood of the seed; hence the NN component discovers all of them.

**Iteration 3** We see that after the second iteration the algorithm has found samples in all $C_1$, $C_3$ and $C_4$. The RandS component discovers these samples. The NN component has not made significant progress in this step since it is still exploring the regions near $C_2$.

**Iteration 4** The addition of the new cluster centres in the previous iteration benefits the NN component most since now it discovers almost all the samples from every cluster. Still, there lies a few samples in the embedding space that seem not to form a cluster and are also well separated from $C_1$, $C_2$, $C_3$ and $C_4$.

**Iteration 5** Since all the possible cluster centers are found the RandS component seems to be struggling but the NN component still probes the neighbourhood of the previously discovered samples inturn finding more positive samples. The search ends after this iteration and next we move on to the final search component of the algorithm.

**Final Search** Using the head trained on data from all the previous iterations, the undiscovered samples' confidence are calculated. The samples having the highest confidence scores are picked up and labelled. This step ensures that all the samples left behind by the RandS and NN components are still discovered.

In this demonstration, we showed that all the positive samples were discovered but real-life datasets are often complicated and suffer from various noises such as mislabelling, domain shift, etc. Hence, not every positive point can be discovered.

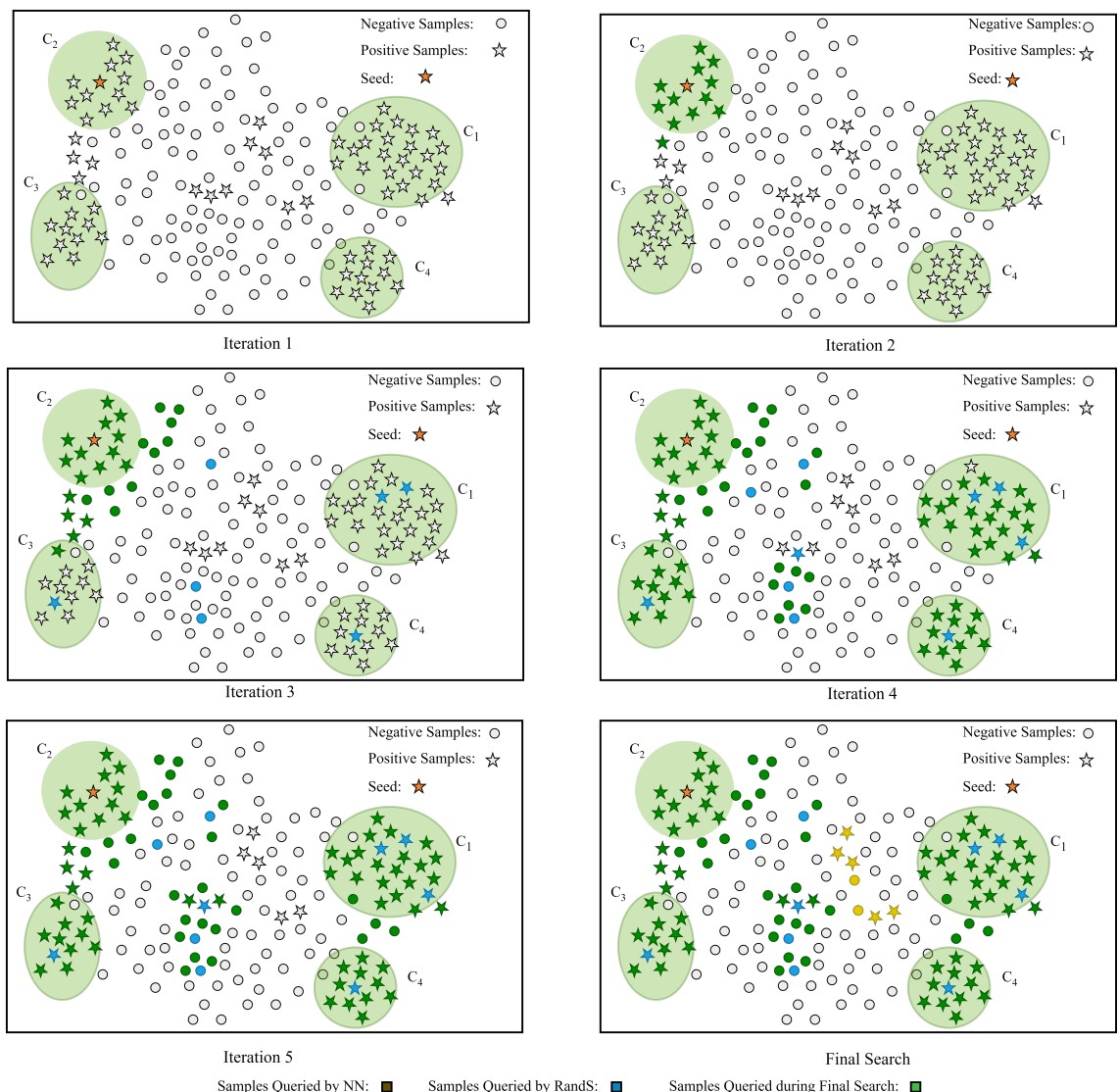

Figure 5: Visual representation of Latent Space, with the corresponding positive and negative samples

# B  APPENDIX: RANDS

In our algorithm for ELSA, the RandS sampler deviates significantly from conventional random samplers. A standard random sampler isn't suitable for our case. The RandS module is designed specifically to identify the most probable random clusters. This section provides an in-depth explanation of how the RandS module operates within the algorithm.

The process begins by randomly selecting 'l' samples from the dataset, ensuring their confidence surpasses a defined threshold. Subsequently, a nearest neighbor search is conducted across the entire dataset. For each 'l' sample, the nearest neighbor (excluding itself) is identified. These 'l' neighbors are then organized based on their distances from the corresponding 'l' points. The algorithm selects the first 'b' points with the smallest distance from their respective 'l' points.

This selection methodology is grounded in the observation that multiple samples from 'l' might belong to a single cluster or be neighbors to each other. Therefore, including all of them would be inefficient since the nearest neighbor component can uncover these relationships. This approach

compels the RandS component to exclusively select points that are distant from one another. Consequently, this orthogonalizes the two search components, as depicted in Figure 6. The illustration demonstrates how these components collaborate to identify positive samples, in stark contrast to the conflicting search dynamics between a baseline random search and the nearest neighbor component.

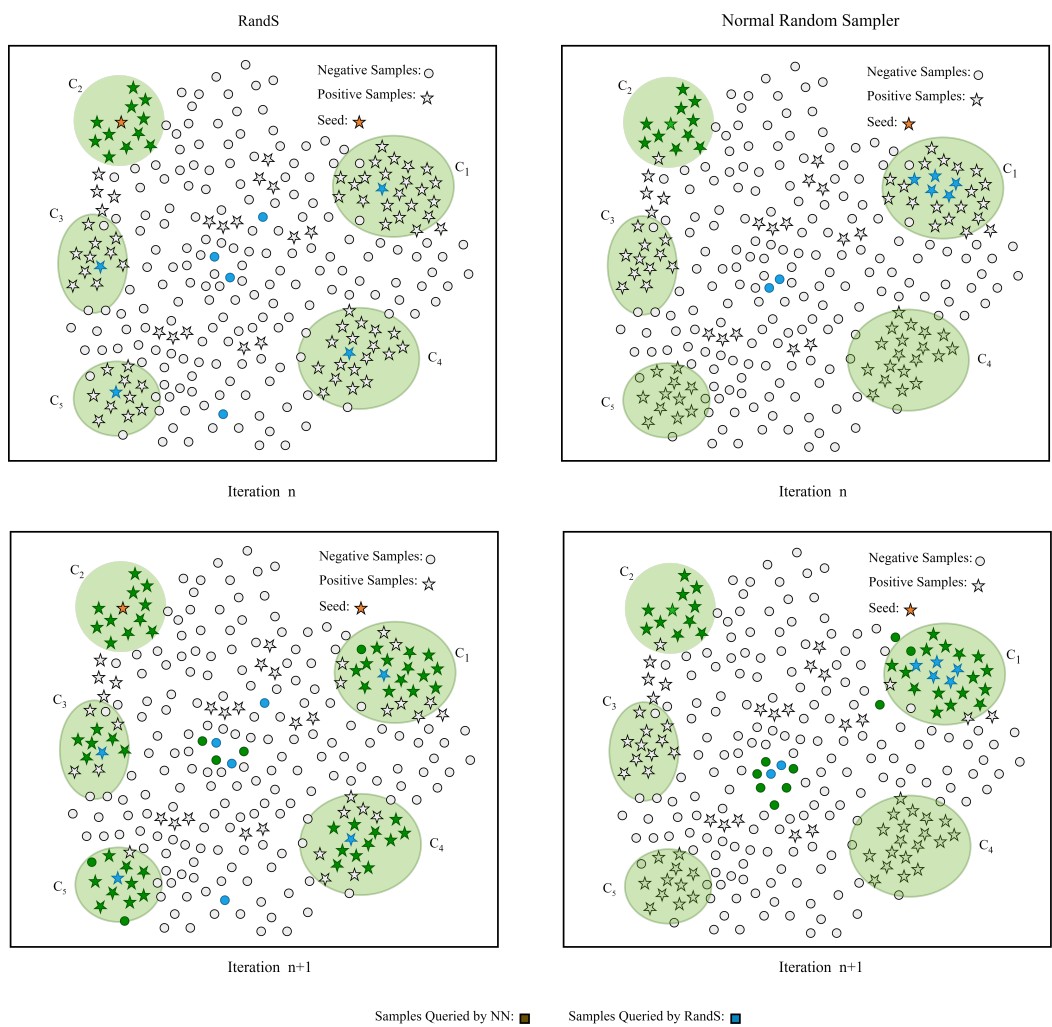

Figure 6: Working of RandS vs. baseline Random Search

## C APPENDIX: HYPERPARAMTERS

In this section we discuss the impact of various hyperparameters on the performance of the algorithm. The proper choice of this hyperparametrs will lead the end user to use this algorithm in a more efficient manner. For all the experiment in this section we run the algorithm using the representation generated by the VICReg backbone. We run our algorithm with the same 50 classes and that we used in (tables 2, 3, 4) and take the median statistics across five independent runs for every single class. We keep the number of seeds fixed to five.

**Nearest Neighbour Search Radius $a$ :** The hyperparameter $a$ is one of the most import hyperparameters that must be set at specific values for the algorithm to work well. This parameter controls how many samples we select from the nearest neighbour sampler hence by increasing the value of a we can effectively increase the radius of the search. This would impact the discovery rate positively but the overall labelling efficiency will go down. From our study, setting the value of a equal to 50 is

a really good trade-off between the labelling efficiency and discovery rate. From table 8 and figure 7 we can clearly conclude the negatve correlation between the two metrics when various values of the hyperparameter $a$ are used.

| a | Labelling Efficiency | Discovery Rate |
|---|---|---|
| 10 | 0.151 | 0.448 |
| 25 | 0.149 | 0.725 |
| 50 | 0.116 | 0.772 |
| 100 | 0.092 | 0.793 |
| 200 | 0.073 | 0.796 |

Table 6: Labelling Efficiency and Discovery Rate for various values of hyperparameter $a$.

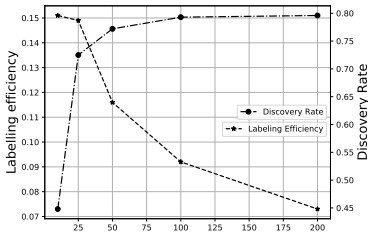

Figure 7: Labelling Efficiency and Discovery Rates vs. a. Labelling Efficiency shows negative correlation to the Discovery Rate.

**Active Learning Sampler Parameter** $\alpha$ :This parameter controls the confidence threshold of the selected samples from the sampler. Since we are using a MLP layer as a head which computes the confidence of a specific sample being positive or not, a threshold needs to be set such that only samples with a higher confidence level are selected for the next stages. In this work $\alpha$ is used in two places, (a) RandS sampler and (b) Final Search. For the sake of completeness, we have explored the effect of alpha in both places. We call the alpha used in case (a) as $\alpha_1$ and the latter as $\alpha_2$. Since $alpha_2$ is at the Final Search stage the end user can choose to completely ignore this parameter by only labelling the most number of confident samples depending on the labelling budget.

| $\alpha_1$ | Labelling Efficiency | Discovery Rate |
|---|---|---|
| 0.2 | 0.054 | 0.840 |
| 0.5 | 0.085 | 0.813 |
| 0.7 | 0.102 | 0.792 |
| 1 | 0.125 | 0.773 |
| 1.5 | 0.137 | 0.721 |
| 2 | 0.155 | 0.727 |

Table 7: Variation in labelling efficiency and discovery rate for change in the hyperparameter $\alpha_1$ a.

| $\alpha_2$ | Labelling Efficiency | Discovery Rate |
|---|---|---|
| 25 | 0.111 | 0.766 |
| 50 | 0.124 | 0.766 |
| 100 | 0.123 | 0.752 |

Table 8: Labelling Efficiency and Discovery Rates vs. a. Labelling Efficiency shows negative correlation to the Discovery Rate.

**RandS Parameter l** : ELSA has a parameter l that determines how many points will be sampled from the active learning head to calculate the cluster centres. l is more of a abstract hyper-parameter but from our studies in table 9 we find that the working of the algorithm is not very sensitive to the choice of l.

| l | Labelling Efficiency | Discovery Rate |
|---|---|---|
| 25 | 0.111 | 0.766 |
| 50 | 0.124 | 0.766 |
| 100 | 0.123 | 0.752 |

Table 9: Variation in labelling efficiency and discovery rate for change in the hyperparameter l

# D    APPENDIX: ERROR ANALYSIS

In the entire work, the oracle is considered perfect. For our evaluation and experiments, we have used the labels from the dataset as the oracle's output. Most real-life active learning-based systems might not have a perfect oracle, introducing noise to the entire search algorithm. Active learning, in

general, is very sensitive to noise. Moreover, a noisy sample can often mislead the entire algorithm towards the wrong direction since we are working with very few positive samples. Hence, proper analysis of noise and the performance of ELSA is required. To simulate noise, we randomly flip $p$(in percentage) of labels during the labelling process, introducing noise to the system. From our study in table 10 and figure 8, we find that although with the introduction of noise, the discovery rate is not impacted much, the labelling efficiency goes down quite aggressively as the noise increases. From this, it can be concluded that ELSA has a small labelling efficiency for noisy oracles but still manages to discover the most positive samples. This problem can solved by using recently developed active learning-based algorithms that are specifically designed for noisy oracles is in cite Zhang & Chaudhuri,Algan & Ulusoy (2021), Wei et al. (2022).

| p(%) | Labelling Efficiency | Discovery Rate |
|------|---------------------|----------------|
| 2.5  | 0.0817              | 0.741          |
| 5    | 0.0491              | 0.748          |
| 10   | 0.0218              | 0.750          |
| 15   | 0.0186              | 0.728          |

Table 10: Variation of performance metrics with introduction of noise.

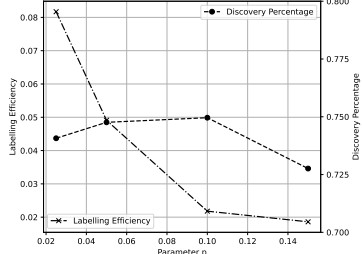

Figure 8: Variation of performance metric with introduction of noise.

## E    APPENDIX: EVALUATION ON FOODS-101 DATASET

The Foods-101[5] dataset consists of approximately 100,000 samples of various food items with 101 classes. Each class consists of 1000 samples in the dataset. We train a VICReg backbone on the dataset using the exact training schedule followed by the authors in VICReg. We test our algorithm on this new dataset using the newly trained self-supervised backbone. The algorithm's performance can be referred to in table 11. The experiment structure is identical to tables 2 and 3.

| Seeds | Labelling Efficiency | Discovery Percentage |
|-------|---------------------|----------------------|
| 1     | 0.1509              | 0.629                |
| 2     | 0.153               | 0.674                |
| 5     | 0.1533              | 0.725                |
| 10    | 0.1406              | 0.765                |
| 50    | 0.1169              | 0.838                |

Table 11: Evaluation of ELSA on Foods-101 dataset, with VICReg backbone.

## F    APPENDIX: COMPONENTS OF ELSA

We have described in appendix A and section 4 how the different components of ELSA work together to discover the positive samples from the given unlabelled dataset. In this section, we compare the per iteration sample discovery of a simple nearest neighbour search with ELSA and its two components in figure 9 for five different classes. We can see how the RandS and NN components outperform a simple nearest neighbour search. The graphs are plotted by taking median statistics across five independent runs with the number of seeds fixed to five and the backbone being VICReg.

---

[5]https://data.vision.ee.ethz.ch/cvl/datasets_extra/food-101/

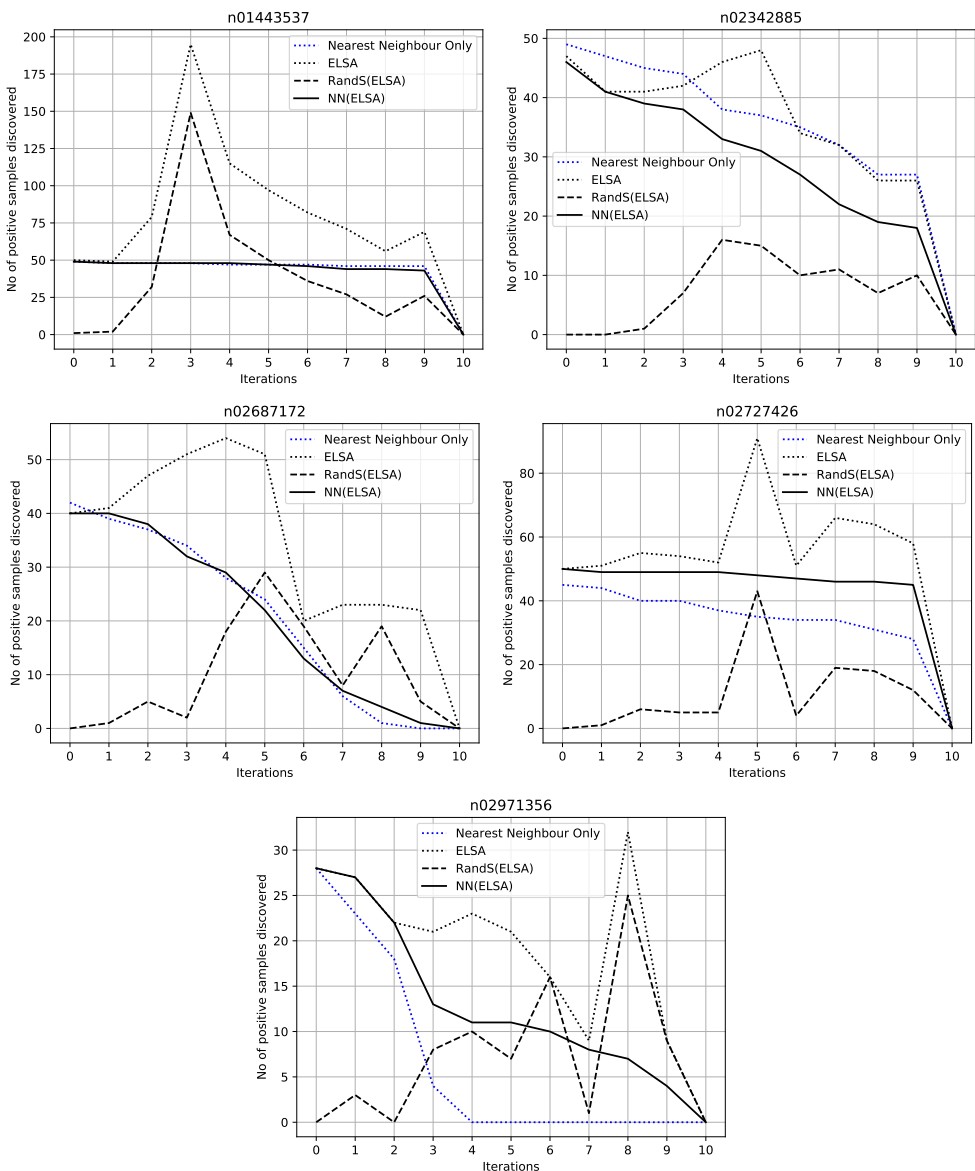

Figure 9: Iteration-wise analysis of RandS and NN in comparison to nearest neighbour baseline

