# OpenReview forum: "Explorative Latent Self-Supervised Active Search Algorithm (ELSA)"
_ICLR.cc/2024/Conference — Submitted to ICLR 2024_

### Official Review · Reviewer_HU18 · 2023-10-26

**Soundness:** 2 fair
**Presentation:** 1 poor
**Contribution:** 1 poor
**Rating:** 1
**Confidence:** 4

**Summary:**

This paper discussed an efficient latent space based self-supervised active learning search. Typically, it focuses on positive class labeling. The method includes three components: a)an iterative Nearest Neighbor Search, b)a Sophisticated Random Sampler, c)a Linear Head powered by Active Learning, Some experiments are done to show the method works to some extent.

**Strengths:**

The paper considers a self-supervised active learning search, which seems to be a potential solution to reduce labeling effort.

**Weaknesses:**

(1)	The novelty of the work is very small as both active learning and self-supervised learning are well known.
(2)	The solution that this paper takes is quite common, including nearest neighbor search and random sampler.
(3) It looks like the pape combines several existing methods into one piece without motivating the method well and explaining why they are combined.

(4) The paper is poorly written without an official problem definition.

**Questions:**

(1) The contribution is not clearly. For instance why nearest neighbor search would be listed as contribution since it is just a common method. Similarly , the random sampling method is also a very routine method.

(2) The method lacks theoretical support. For instance, what is the error bound ? How does the method perform under noisy setting ?

(3) The experimental evaluations are weak and not sufficient.

---

> ### Author Response · Authors · 2023-11-17
>
> ### **Weakness**
> 1. *The novelty of the work is very small as both active learning and self-supervised learning are well known.*
>
> Thank you for your comments. Both active learning and self-supervised learning techniques are used in developing ELSA. ELSA is more of a search algorithm than an active learning technique. The primary motivation behind the development of an algorithm like ELSA is to ease the task of manual labelling.
>
> 2. *The solution that this paper takes is quite common, including nearest neighbor search and random sampler.*
>
> We have used an ordinary nearest neighbour search routine for this task, which forms one of our components, but our random sampler is unique. It differs from the usual random search because it tries to find the most probable cluster centres from the dataset. We have shown in detail in Appendix B how RandS differs from a vanilla random sampler. Moreover, we also provide the intuition behind why RandS works so well in Appendix B. RandS is designed to search orthogonally to the nearest neighbour sampler, making the two work independently and symbiotically.  Appendix F shows how adding the RandS improves the search compared to the nearest neighbour search.
>
> 3. *It looks like the pape combines several existing methods into one piece without motivating the method well and explaining why they are combined.*
>
> We have added Appendix A, where we provide a detailed description of the algorithm and the interplay of its components. We have also added Section 4.4 in the paper in order to motivate the readers to the problem statement and give a basic intuition on the working of ELSA. Both sections describe why we need all the components used to design the algorithm.
>
> 4. *The paper is poorly written without an official problem definition.*
>
> We have made some changes following the comments of the reviewer. We have added section 4.4,  Appendix A, Appendix B and Appendix F to make the working of the algorithm and the problem statement more straightforward. If the reviewer has some additional comments, we are open to them.
>
>
> ### **Questions**
> 1. *The contribution is not clearly. For instance why nearest neighbor search would be listed as contribution since it is just a common method. Similarly , the random sampling method is also a very routine method.*
>
> Our random sampler differs from the vanilla one, as described in Appendix B. We have not claimed the nearest neighbour search as our contribution rather, it has been described and formalised for the sake of completeness.
>
> 2. *The method lacks theoretical support. For instance, what is the error bound ? How does the method perform under noisy setting ?*
>
> ELSA works on the embedding space of a self-supervised algorithm. The mathematical formation of these spaces is still an active field of research; hence, providing any theoretical bound of error for the algorithm will be very difficult due to the sheer complexity of such spaces. We believe this to be beyond the scope of our study and hence have not explored the theoretical error bound. Nevertheless, we have added experiments in the appendix (Appendix D) to show how our algorithm performs under noisy settings. ELSA works well even in moderately noisy settings, but the labelling efficiency sufferers as we add more noise. The discovery rate remains approximately constant to the overall noise added. If the reviewer wants any other theoretical justification, we will be happy to answer them.
>
> 3. *The experimental evaluations are weak and not sufficient.*
>
> Following the suggestions from all the reviewers, we have added more experiments and analysis in Appendix C, E and F. We would like to ask the reviewer, for any additional experiments that they might need. We are open to any further suggestions.

---

> > ### Comment · Reviewer_HU18 · 2023-11-22
> > **Final comments**
> >
> > I have read the feedback from the authors and I still have the concerns from novelty, lack of theory to support, insufficient experiments as well as presentation issues. Thus, I maintain my review score.

---

### Official Review · Reviewer_4JSg · 2023-10-30

**Soundness:** 3 good
**Presentation:** 1 poor
**Contribution:** 2 fair
**Rating:** 5
**Confidence:** 2

**Summary:**

This paper proposes an active learning-based labeling assistant algorithm, called ELSA. The author clarifies that ELSA can help reduce the time required to label samples by orders of magnitude compared to manual labeling in large datasets dominated by many negative samples. The paper provides insights into the interplay of these components and their collective impact on search efficiency. Finally, the paper also presents proof-of-concept empirical experiments to corroborate the theoretical results.

**Strengths:**

Although I haven't thoroughly read most of the technical proofs, the results appear sound and technically correct.

**Weaknesses:**

The paper is in general easy to follow and well-structured. There are some interesting theoretical guarantees, which seem simple and effective. Nevertheless, I have the following concerns:

1. Not enough empirical evaluations.  it necessary to evaluate other state-of-the-art tabular benchmarks in Table 5.
2. Novelty and limitations. The theoretical justification is interesting but the novelty in the method itself is slightly incremental, and the proposed algorithm seems based on a simple modification.

**Questions:**

The paper has a few English typos in different places. The setting studied in the paper is quite classical. The novelty is harder to judge for me (see my comment in the "weaknesses" above) but the method and algorithm proposed seem quite classical. However, because of my unfamiliarity with the related works, this is a low-confidence review.

I could not find the code to check reproducibility.

---

> ### Author Response · Authors · 2023-11-17
>
> ### **Weakness**
> 1. *Not enough empirical evaluations. it necessary to evaluate other state-of-the-art tabular benchmarks in Table 5.*
>
> Thank you for your comments. Table 5 already includes eight different state-of-the-art self-supervised methods on which we have benchmarked our algorithm.  It would be helpful if the reviewer could elaborate on this statement: *"necessary to evaluate other state-of-the-art tabular benchmarks"*; if the reviewer wants us to run more benchmarks, their suggestions are welcome.
>
> 2. *Novelty and limitations. The theoretical justification is interesting but the novelty in the method itself is slightly incremental, and the proposed algorithm seems based on a simple modification.*
>
> The novelty of the method lies in connecting the different search components. We have used two main components in our algorithm, one of which is the standard nearest neighbour search, the other one being RandS. We claim RandS as one of our main contributions to this paper. We have described in detail in Appendix A how RandS, along with the NN component, can maximise the search. In addition, we have also described in Appendix B why a sophisticated random sampler like RandS is required for this purpose and why the usual random sampler would fail in this scenario. There are multiple works on automatizing the task of labelling in the literature, some of which are described in the paper and how they are different from our work; moreover, we focus on a more general dataset like ImageNet and Food101 to benchmark and test our algorithm. We have also shown from our analysis in Appendix D how the two components of ELSA work orthogonal to each other and maximise the search without affecting each other.
>
> 3. "I could not find the code to check reproducibility"
>
> All our source codes are in the supplementary material along with the analysis notebooks and log files. We plan to link the official Github repository in the camera-ready version.

---

> > ### Comment · Reviewer_4JSg · 2023-11-22
> >
> > I have read the feedback from the authors and I still maintain my review score.

---

### Official Review · Reviewer_3EJ7 · 2023-11-04

**Soundness:** 2 fair
**Presentation:** 2 fair
**Contribution:** 3 good
**Rating:** 5
**Confidence:** 3

**Summary:**

The paper presents a new approach to active learning-based labeling called ELSA, or the Explorative Latent Self-Supervised Active Search Algorithm. ELSA is designed to address the challenges of imbalanced datasets and limited positive samples in computer vision applications. The authors demonstrate that ELSA can achieve high levels of precision and accuracy even with just a few positive samples, making it a cost-effective and practical alternative to manual annotation. The paper also outlines the three fundamental components of ELSA and provides experimental results to support its effectiveness. Overall, the paper's contributions include a novel approach to active learning-based labeling, a detailed description of the ELSA algorithm, and empirical evidence of its effectiveness in computer vision applications.

**Strengths:**

1. The authors propose a novel active learning-based labeling method which combines a Nearest Neighbour search module, A Random Sampler and a classification head, achieves orders of magnitude superior performance.

2. The authors provide empirical evidence of the effectiveness of ELSA in computer vision applications, using several benchmark datasets and evaluation metrics.

3. The paper also includes a thorough analysis of the results, discussing the strengths and limitations of the approach and comparing it to other state-of-the-art methods.

**Weaknesses:**

1. Lack of analysis of the impact of hyperparameters such as “a” in the NEAREST NEIGHBOUR SEARCH: The authors do not provide a detailed analysis of the impact of hyperparameters on the performance of ELSA. This is an important aspect of the algorithm, as the choice of hyperparameters can have a significant impact on its effectiveness. A more thorough analysis of the impact of hyperparameters would help to identify the optimal settings for different datasets and applications.

2. The meanings of some symbols such as $L_e$ and $d_r$ in Section3.1 have not been provided with sufficient clarity.

**Questions:**

1. How sensitive is ELSA to the choice of hyperparameters? Can you provide a more detailed analysis of the impact of hyperparameters on the performance of the algorithm? This would help to identify the optimal settings for different datasets and applications.

2. How might ELSA be extended or modified to address other challenges in active learning-based labeling, such as the presence of noisy or mislabeled data?

---

> ### Author Response · Authors · 2023-11-17
>
> ### **Weakness**
> 1. *Lack of analysis of the impact of hyperparameters such as “a” in the NEAREST NEIGHBOUR SEARCH: The authors do not provide a detailed analysis of the impact of hyperparameters on the performance of ELSA. This is an important aspect of the algorithm, as the choice of hyperparameters can have a significant impact on its effectiveness. A more thorough analysis of the impact of hyperparameters would help to identify the optimal settings for different datasets and applications.*
>
> Thank you for your comments. Following the suggestions from the reviewers, we have done an in-depth analysis of several hyperparameters of our algorithm. We have added the hyperparameter analysis to the appendix of our paper (Appendix C).
>
> 2. *The meanings of some symbols such as $L_e$ and $d_r$ in Section3.1 have not been provided with sufficient clarity.*
>
> $L_e$ is the labelling efficiency, i.e. the ratio of the number of positive samples discovered to the total number of samples labelled during the search process.
> $d_r$ is the discovery rate, i.e. the ratio of the number of positive samples discovered by the search algorithm to the total number of positive samples present in the dataset. To give a better understanding to the readers, we have updated section 3.1
> to provide better clarity.
>
> ### **Questions**
> 1. *How sensitive is ELSA to the choice of hyperparameters? Can you provide a more detailed analysis of the impact of hyperparameters on the performance of the algorithm? This would help to identify the optimal settings for different datasets and applications.*
>
> ELSA is robust when it comes to the choice of hyperparameters. Please see Appendix C for a detailed analysis.
>
> 2. *How might ELSA be extended or modified to address other challenges in active learning-based labeling, such as the presence of noisy or mislabeled data?*
>
> We have added a section for noise analysis for ELSA, which can be found in the appendix (Appendix D). From our study, we see that ELSA works even in moderately noisy settings, but the labelling efficiency of the algorithm suffers as more noise is added to the algorithm. The discovery rate remains approximately constant to the overall noise that is added. The issue can be tackled using a more sophisticated active learning sampler that works well in noisy settings. Due to the sheer simplicity of the algorithm, changing the active learning component to a more sophisticated one should not be very difficult.

---

> > ### Comment · Reviewer_3EJ7 · 2023-11-21
> > **Reply to authors**
> >
> > Thank the authors for the additional elaboration on the details of the paper, which has clarified the concepts such as labelling efficiency and discovery rate. Furthermore, the experiments on algorithm hyperparameters in Appendix C are comprehensive, providing a more thorough explanation of the algorithm's robustness and the correctness of hyperparameter selection. Here are some additional suggestions and concerns.
> > 1. The paper could benefit from a clearer structure. For instance, in the beginning of Section 4, the explanation of the algorithm's different components is a bit messy. A suggested improvement is to organize it with Samper first, followed by Random Search, and then NN for better coherence.
> > 2. In Section 4.4, the author could strengthen the explanation of why the algorithm works by referencing previous works or providing experimental proof. To make the argument more convincing, it's suggested to back up claims with references. Also, consider using actual data points instead of Figure 1 for a more persuasive visualization.
> > 3. There are some symbol consistency problems in this paper. For example, in Algorithm 1, $\mathcal M_{ij}$ should be written as $\mathcal M_{ij} = MSE(g(d_i), g(\Lambda_j))$. It's important to use the same symbols for the same concepts, like using $I_i$ in Section 3.1 but $d_i$ in Section 4.1.
> > 4. Although I'm not very familiar with the related works, the discussion of algorithmic time and space complexity in Section 5 may not be crucial for this task.
> > 5. The paper and appendices experiment with hyperparameters and embedding spaces to validate the effectiveness of each component, but it lacks a crucial aspect—a comparison with popular existing methods. It's important to show how the algorithm performs compared to other well-known techniques.
> >
> > The authors conduct extensive analyses, including experiments on various hyperparameters and architectures for the proposed algorithm. However, considering some shortcomings in the writing aspect of the paper, and the insufficient reasoning behind why the algorithm is effective, as well as the lack of comparisons with more state-of-the-art (SOTA) methods in the experiments section, it is recommended to assign a rating marginally below the acceptance threshold.

---

### Meta-Review · Area_Chair_PU9t · 2023-12-06

**Metareview:**

The submission presents an active learning algorithm for use with computer vision models, in particular where negative class labels dominate the distribution.  The proposed algorithm combines several components (nearest neighbor search, random search, and linear head based score),  which are all evaluated in an ablation study.

I do not recommend accepting the paper in its current form. The paper does a convincing job of justifying the design decisions made for the proposed approach, but it critically is lacking comparisons to other state-of-the-art active learning approaches (as highlighted by the reviewers).  Adding these comparisons will significantly increase the quality of the submission and should be addressed before publication.

**Justification For Why Not Higher Score:**

Lack of comparisons to other baselines in a primarily empirical paper is a critical flaw.

**Justification For Why Not Lower Score:**

N/A

---

### Decision · Program_Chairs · 2024-01-16

Reject